# Me-Momentum: Extracting Hard Confident Examples from Noisily Labeled Data

## Abstract

Examples that are close to the decision boundary—that we term *hard examples*, are essential to shaping accurate classifiers. Extracting confident examples has been widely studied in the community of learning with noisy labels. However, it remains elusive how to extract hard confident examples from the noisy training data. In this paper, we propose a deep learning paradigm to solve this problem, which is built on the *memorization effect* of deep neural networks that they would first learn *simple patterns*, i.e., which are defined by these shared by multiple training examples. To extract hard confident examples that contain non-simple patterns and are entangled with the inaccurately labeled examples, we borrow the idea of momentum from physics. Specifically, we alternately update the confident examples and refine the classifier. Note that the extracted confident examples in the previous round can be exploited to learn a better classifier and that the better classifier will help identify better (and hard) confident examples. We call the approach the "***Momentum* of *Me*morization**" (Me-Momentum). Empirical results on benchmark-simulated and real-world label-noise data illustrate the effectiveness of Me-Momentum for extracting hard confident examples, leading to better classification performance.

## 1 Introduction

As training datasets are growing big while accurately labeling them is often expensive or sometimes even infeasible, cheap datasets with label noise are ubiquitous in many real-world applications. Without any care, label noise will degenerate the performance of learning algorithms, especially for these based on deep neural networks (Zhang et al., 2017). Learning with noisy labels (Angluin & Laird, 1988) aims to reduce the side-effect of label noise and therefore has become an important topic in machine learning.

Existing methods for learning with noisy labels can be divided into two categories: algorithms that result in *statistically consistent* or *inconsistent* classifiers. Methods in the first category intent to design *classifier-consistent* algorithms (Zhang & Sabuncu, 2018; Kremer et al., 2018; Liu & Tao, 2016; Scott, 2015; Natarajan et al., 2013; Goldberger & Ben-Reuven, 2017; Patrini et al., 2017; Thekumparampil et al., 2018; Yu et al., 2018; Liu & Guo, 2020; Xu et al., 2019), where classifiers learned by using the noisy data will statistically converge to the optimal classifiers defined by clean data. However, these methods rely heavily on the *noise transition matrix* (Liu & Tao, 2016; Patrini et al., 2017; Xia et al., 2019). In real-world applications, it is hard to learn the *instance-independent* noise transition matrix (Cheng et al., 2020). To be free of estimating the noise transition matrix, methods in the second category employ heuristics to reduce the side-effect of label noise (Yu et al., 2019; Han et al., 2018b; Malach & Shalev-Shwartz, 2017; Ren et al., 2018; Jiang et al., 2018; Ma et al., 2018; Kremer et al., 2018; Tanaka et al., 2018; Reed et al., 2015; Han et al., 2018a; Guo et al., 2018; Veit et al., 2017; Vahdat, 2017; Li et al., 2017; 2020). These methods were reported to empirically work well, especially in the setting of *instance-dependent* label noise.

One promising direction in the second category is to extract examples with clean labels—*confident examples*— (Northcutt et al., 2017; Cheng et al., 2020; Chen et al., 2019; Malach & Shalev-Shwartz, 2017; Han et al., 2018b; Yu et al., 2019; Thulasidasan et al., 2019; Nguyen et al., 2020; Dredze et al., 2008; Liu & Tao, 2016; Wang et al., 2017; Ren et al., 2018; Jiang et al., 2018). The idea is that compared with the original noisy training data, the extracted examples are less noisy and thus

Figure 1: The illustration of the influence of hard (confident) examples in classification. Circles represent positive examples while triangles represent negative examples. Green and blue denote examples with accurate labels while red presents examples with incorrect labels. Blank circles and triangles represent unextracted data. (a) shows an example of classification with clean data. (b) shows noisy examples, especially those close to the decision boundary, will significantly degenerate the accuracy of classifier. (c) shows confident examples help learn a fairly good classifier. (d) shows that hard confident examples are essential to train an accurate classifier.

will lead to a classifier with better performance. Given only noisy data, state-of-the-art methods exploit the *memorization effect* (Zhang et al., 2017; Arpit et al., 2017) to extract confident examples. The memorization effect will enable deep neural networks to first learn patterns that are shared by majority training examples. As clean labels are of majority in each noisy class (Natarajan et al., 2013; Xiao et al., 2015), deep neural networks would therefore first fit training data with clean labels, and then gradually fit the examples with incorrect labels (Chen et al., 2019). Therefore, early stopping (Li et al., 2020; Song et al., 2019) and the small loss trick (Jiang et al., 2018; Han et al., 2018b; Yu et al., 2019) can be used to exploit confident examples.

Examples that are close to the decision boundary are called *hard examples*. As illustrated in Figure 1, hard (confident) examples play an important role in shaping the decision boundary. It has also been widely studied in the traditional classification problem that hard examples are essential to train accurate classifiers (Vapnik, 2013; Bengio et al., 2009; Huang et al., 2010; He et al., 2018). Notwithstanding the importance of hard confident examples, none of the existing methods studies how to extract hard confident examples from noisy data. Note that extracting hard confident examples is non-effortless. Since hard examples are often of a small proportion and contain less discriminative information compared with the easy ones (these that are far away from the decision boundary), they are often entangled with inaccurately labeled examples in the procedures of extraction.

In this paper, by alternately updating the confident examples and refining the classifier, we propose a deep learning paradigm that is able to extract hard confident examples from the noisy training data, leading to better classification performance. Specifically, the idea is similar to the usage of momentum from physics. As stated in the statistical learning theory, with better training data, better classifier can be obtained (Mohri et al., 2018). We can then think of the classifier as a particle traveling through the hypothesis space, getting acceleration from the confident data. Classifiers with better performance can be achieved by properly exploiting the previously extracted confident examples. This is similar to the momentum trick used in optimization that previous gradient information can be used to escape local minimum and achieve fast convergence rates (Sutskever et al., 2013)[1]. At a high level, the proposed method is built on the memorization effect of deep neural networks and on the intuition that better confident examples will result in a better classifier and that a better classifier will identify better confident examples (and hard confident examples). The proposed method is therefore called the Momentum of Memorization (Me-Momentum).

We conduct experiments to show the effectiveness of the proposed Me-Momentum on noisy versions of *MNIST*, *CIFAR10*, *CIFAR100*, and a real-world label noise dataset *Clothing1M*. Specifically, on *MNIST* and *CIFAR*, we generate class- and instance-dependent label noise and visualize the extracted hard confident examples, which justifies why Me-Momentum consistently outperforms the baseline methods. Codes will be available online.

## 2   ME-MOMENTUM

In this section, by specifying the proposed method of *momentum of memorization* (Me-Momentum; summarized in Algorithm 1), we would like to detail how to accomplish extracting hard confident

---

[1]In optimisation, the parameter vector can be thought of as a particle traveling through the parameter space, getting acceleration from the gradient of the loss. The momentum trick demonstrated that gradient in previous update can help escape local minimum and achieve fast convergence rates.

---

**Algorithm 1** Me-Momentum

---

**Input**: Noisy training data, noisy validation data, iteration number $N_{\text{inner}}$ and $N_{\text{outer}}$;
**Output**: Extracted confident examples and the classifier;
1: **Initialize** a classifier $f_0$ by using the noisy training data and early stopping; //memorization effect
**for** $i = 1, \ldots, N_{outer}$ **do**
    **for** $j = 1, \ldots, N_{inner}$ **do**
        2: **Update** the extracted confident examples;
          //i.e., the training examples whose noisy labels are identical to the ones predicted by $f_{j-1}$
        3: **Obtain** the classifier $f_j$;
          //initialize the network by using the parameters of $f_{j-1}$ and train it by employing confident examples; the classifier $f_j$ will be chosen with the highest noisy validation accuracy throughout the training procedure
        4: **Break** and output $f_{j-1}$ if the highest validation accuracy is non-increasing in the loop;
    **end**
    5: **Re-initialize** a classifier $f_0$;
        //randomly initialize the network and train it by using confident examples; the classifier $f_0$ will be chosen with the highest noisy validation accuracy throughout the training procedure
    6: **Break** and output $f_{j-1}$ if the highest validation accuracy is non-increasing in the loop;
**end**

---

examples and boosting the classification performances. At the high level, by alternately updating the confident examples and refining the classifier, Me-Momentum fulfils a positive cycle that better confident examples will result in a better classifier and that a better classifier will identify better confident examples. Specifically, Me-Momentum has two loops, i.e., an inner loop and an outer loop. In the inner loop, Me-Momentum alternates update of the confident examples and classifier (Steps 2 and 3). However, the inner loop continually refines a classifier and thus depends heavily on the initialization of the classifier (Step 1). It may lead to the memorization of noisy labels and the inferiority of sample-selection bias. To handle this problem, the outer loop re-initialize the classifier (Step 5) while maintains the previously extracted confident examples.

There are some points to be clarified for the proposed Algorithm 1:

    Q1. How to initialize a good classifier in Step 1?

    Q2. How to extract confident examples in Step 2?

    Q3. How to validate the learned classifiers in Steps 3 and 5 without a clean validation set?

    Q4. Why the proposed method is called Me-Momentum?

    Q5. Why hard confident examples can be extracted?

To answer the *first question*, we would like to mention that the aim of the initialization in Step 1 is to initialize a good classifier for the positive cycle: a better classifier will identify better confident examples and better confident examples will result in a better classifier. A good candidate should have a fairly high classification accuracy, e.g., higher than random guessing. Otherwise, the positive cycle cannot be evoked. Fortunately, the initialization can be made by exploiting the memorization effect of deep neural networks that they would first fit clean data (Arpit et al., 2017; Zhang et al., 2017). Note that this memorization effect is independent of training optimization or network backbones (Arpit et al., 2017). Specifically, we use the early stopping trick. For easy understanding, we would like to introduce a definition of *high-peak*. A noisy validation accuracy at the $i$-th epoch is called an $i$-th high-peak if it achieves the highest accuracy in the epoch range $\{1, \ldots, i\}$. Assume the $i$-th and $j$-th high-peaks occur next to each other, having noisy validation accuracies of $a$ and $b$, respectively. The training early stops if $(b - a)/(j - i) \leq \tau$, where $\tau$ is a hyper-parameter. In the experiments, we set $\tau = 0.1$, which empirically works well across all datasets. In Appendix A, we compare the difference between the early stopping method and the traditional validation method; We also study the sensitivity of the hyper-parameter.

The answer to the *second question* is closely related to the memorization effect. Note that the classifier initialized in Step 1 would fit the clean data well but not the incorrectly labeled data because of the memorization effect and early stopping. Therefore, we can treat the training examples whose noisy labels are identical to the ones predicted by the classifier obtained in Step 1 as confident examples. This also applies for the classifiers in Step 3 to extract confident examples, which are

iteratively trained by employing the updated confident data. Note that there are some other feasible methods to extract confident examples, e.g., extract those who have a large class posterior.

In Step 3, we aim to learn a better classifier compared with the one in the previous round. This can be achieved because of two reasons: (1) we initialize the network with the parameters of the classifier learned in the previous round; (2) we have a better set of confident examples as the training sample. This starts the positive cycle that better confident examples will result in a better classifier and that a better classifier will identify better confident examples. The *third question* is essential for identifying the classifiers in the cycle. Note that the accurately labeled examples are always assumed to be dominant in each class in the community of learning with noisy labels (Natarajan et al., 2013; Liu & Tao, 2016; Han et al., 2018b). Otherwise, the true class label cannot be identified by only exploiting the noisy data. This assumption implies that the performance on the noisy validation set (split from the noisy training set) and these on test set are positively correlated. The noisy validation set could be used as a surrogate to validate the classifiers if no clean validation set is available. We therefore validate the classifiers in Steps 3 and 5 with the highest noisy validation accuracies during the training. Empirical results show that it works well.

To answer the *fourth question*, we would like to first mention that the proposed method heavily relies on the memorization effect of deep neural networks. Specifically, in Step 1, a classifier is initialized by exploiting the memorization effect via early stopping, which is used to identity confident examples. Later, the classifier and confident examples are iteratively refined and updated, respectively, which is the positive cycle we have mentioned before. Note that this cycle also depends on the memorization effect to update confident examples and refine classifiers. Our method is named as *momentum of memorization* (Me-Momentum) because it also uses the trick of momentum to better exploit the memorization effect. Specifically, we can think of the classifier as a particle traveling through the hypothesis space, getting acceleration from the updated extracted confident data. We exploit the previously extracted confident examples to help learn a better classifier, i.e., initializing the network by using the parameters of the classifier trained with the confident examples extracted in the previous round. The impact of the confident examples in a certain round will decay as we continue updating the classifier.

We intuitively answer the *fifth question* by employing Figure 2. Hard confident examples will be progressively extracted also due to the momentum trick, i.e., previously extracted confident examples will help identify hard confident examples. Specifically, in Figure 2(a), we show the noisy training examples, where circles and triangles represent two classes respectively and the red ones denote the examples with incorrect labels. Let the dash line in Figure 2(b) denote a classifier learned by Me-Momentum. We can use it to extract confident examples, e.g., the green circles and blue triangles, according to Step 2 of Algorithm 1. Compared with Figure 1(a), we can see that hard confident examples close to the optimal boundary, i.e., the circle and triangles with red edges, will be extracted. Note that as better and better classifiers will be learned by Me-Momentum, more and more hard confident examples will be extracted. An visualization is shown in Figure 4.

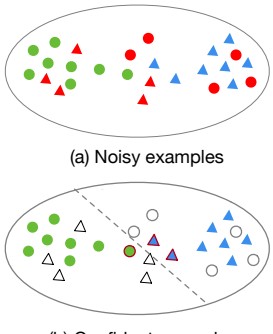

(a) Noisy examples

(b) Confident examples

Figure 2: Illustrative of extracting hard confident examples.

**Relation to existing work:** The strategy of alternatively optimizing the classifier and updating the training examples is not new for dealing with label noise. For example, Joint Optim (Tanaka et al., 2018), Co-teaching (Han et al., 2018b; Yu et al., 2019), and SELF (Nguyen et al., 2020) are similar to ours. Specifically, Joint Optim and Co-teaching update the classifier with one step of stochastic gradient descent while SELF and the proposed method refine the classifier to be optimal with respect to the extracted confident examples. However, existing methods have not focused on extracting hard confident examples and thus are substantially different from this paper because they neglected the importance of avoiding the accumulated error caused by the single initialization of the classifier and the sample-selection bias. Experiments (e.g., Figures 3 and 4) show that Me-Momentum with the outer loop part (i.e., re-initialization of the classifier) contributes significantly to extracting hard confident examples and achieving high label precision. Note that without employing the ensemble technique, SELF is a special case of Me-Momentum with only the inner loop part. We avoid to integrate the ensemble technique to the proposed method for simplicity and clarity.

Me-Momentum is similar to curriculum learning as it also learns from easy to difficult. However, curriculum learning needs a predefined curriculum (sample weighting scheme), e.g., assigning big/small wights for confident/noisy data. If the curriculum is not available, some clean data is required to learn a mentornet to provide a curriculum (Jiang et al., 2018) or a latent variable could be introduced by self-paced learning (Kumar et al., 2010) to learn a curriculum. Differently, Me-Momentum only based on noisy data and does not explicitly learn a curriculum. Me-Momentum also has a similar flavor to active learning which tends to choose and label hard examples to learn from at each iteration. However, for active learning, no label information is available before the data is chosen while Me-Momentum has noisy labels and have to consider the side-effect of label noise.

## 3 EXPERIMENTS

**Datasets:** To verify the effectiveness of the proposed method, we do experiments on datasets with both synthetic and real-world label noise. Specifically, we manually corrupt *MNIST* (LeCun et al., 1998), *CIFAR10*, and *CIFAR100* (Krizhevsky, 2009) with class-dependent label noise and instance-dependent label noise. We detail how to generate class-dependent and instance-dependent label noise in the supplementary material. We employ the real-world noisy dataset *Clothing1M* (Xiao et al., 2015). These datasets have been widely used in studies with noisy labels (Han et al., 2018b; Tanaka et al., 2018; Xia et al., 2019).

For *MNIST*, *CIFAR10*, and *CIFAR100*, we leave out 10% of the noisy training data as noisy validation data. *Clothing1M* contains one million noisy training images, which are crawled from shopping websites, labeling by surrounding text. Almost all existing work uses the $14k$ clean validation data in their experiments. To verify the robustness of the proposed method, we also employ noisy validation data in our experiments. Specifically, $100k$ noisy data are randomly left as noisy validation data and the remaining $900k$ noisy data as the training data.

**Baselines:** Me-Momentum is compared against the following state-of-the-art approaches. (1) Statistically consistent methods: Forward (Patrini et al., 2017), T-revision (Xia et al., 2019), and DMI (Xu et al., 2019); (2) Statistically inconsistent methods: Decoupling (Malach & Shalev-Shwartz, 2017), MentorNet (Jiang et al., 2018), Co-teaching (Han et al., 2018b), Joint Optim (Tanaka et al., 2018), SELF (Nguyen et al., 2020) where MentorNet, Co-teaching and SELF use the idea of extracting confident examples by employing the small loss trick.

**Network structure and optimization:** All the methods are implemented with default parameters by PyTorch v1.5. For the experiments on *MNIST*, *CIFAR10*, and *CIFAR100*, we set $N_{\text{inner}} = 20$, $N_{\text{outer}} = 3$, and follow the settings of T-revision (Xia et al., 2019). Specifically, LeNet-5, ResNet-18 and ResNet-34 networks are used for *MNIST*, *CIFAR10* and CIFAR100 respectively. We use SGD with momentum 0.9, weight decay $10^{-4}$, batch size 128, and an initial learning rate of $10^{-2}$, divided by 10 after the 40-th epoch and 80-th epoch respectively (we fix the learning rate of $10^{-2}$ for the early stopping method). Data augmentation is used with horizontal random flips and $32 \times 32$ random crops after padding 4 pixels on each side. For *Clothing1M*, a ResNet-50 is used. We set $N_{\text{inner}} = 6$ and $N_{\text{outer}} = 3$. To show the effectiveness of the proposed method, we do experiments by randomly initializing the network and pre-training it by employing *ImageNet*, respectively. As the noisy training sample contains a large number of examples, we train the network for 5 epochs and choose the best one with the highest classification accuracy on the noisy validation set. We use SGD with momentum 0.9, weight decay $10^{-3}$, batch size 32, with a learning rate of $5 \times 10^{-3}$, and divided it by 10 at the 3-rd and 5-th round in the inner loop. For each outer loop, the model will be randomly re-initialized (or replaced by a pre-trained one). The learning rate will be reset to $5 \times 10^{-3}$. For data augmentation, all images are resized to $256 \times 256$, horizontal random flipped, and cropped in the middle with a size of $224 \times 224$.

Note that due to the page limit, some complementary experiments to Sections 3.1 and 3.2 and comparison with SELF are put in the Appendix.

### 3.1 VERIFY MOMENTUM OF MEMORIZATION

In Section 2, we discussed that Me-Momentum is implemented by fulfilling the positive cycle that better confident examples will result in a better classifier and that a better classifier will identify

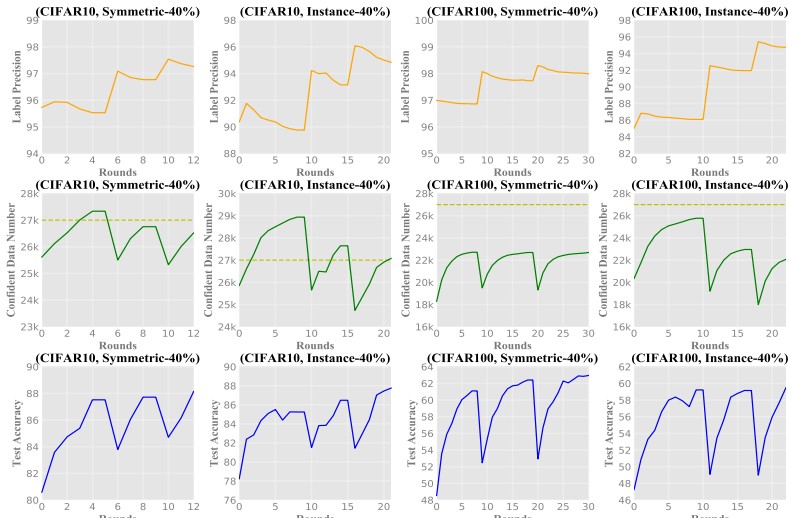

Figure 3: We call one update of the classifier and the extracted confident examples as one round. We illustrate how the label precision of the extracted confident examples, the number of the extracted confident examples, and the classification accuracy of the classifier trained by using the extracted confident examples change during the training of Me-Momentum. We have three distinct peaks in these figures because we have set $N_{\text{outer}} = 3$ and the classifiers are re-initialized in the outer loop. The dash lines in the second row indicate the number of clean labels in the noisy training data.

better confident examples. In this subsection, we will empirically verify this positive cycle, which can be done on the synthetic datasets as we have their ground-truth labels.

In Figure 3, we can see that in the inner loops (e.g., rounds 0-5, rounds 6-9, and rounds 10-12 in the first column of figures represent three inner loops respectively), the classification accuracy generally increases (note that the figures are not smooth because the classifiers are tested on the unseen test data) and the number of extracted confident examples clearly increases (although their label precision decreases slightly). We can also see that in the outer loops (e.g., rounds 0, 6, and 10 in the first column of the figures consist of an outer loop), the classification accuracy clearly increases and the label precision of the extracted confident examples clearly increases (although the number of extracted confident examples decreases slightly). This implies that compared with previous classifiers and extracted confident examples, better ones are obtained, which empirically justifies the positive cycle. Note that the classification accuracy in the outer loop is low because the models are re-initialized.

Figure 3 also shows the importance of the outer loop of Me-Momentum. We can see that the label precision of the extracted confident examples slightly decreases in the inner loops. This is because the deep model gradually memorizes the noisy labels as we continually refine it. This issue can be handled by re-initialize the deep model in the outer loop. Specifically, we can see from Figure 3 that by re-initializing the model in the outer loop, the label precision of the extracted confident examples increases significantly. Although the number of the extracted confident examples decreases, the overall quality of the extracted confident examples is increasing as evidenced by the increase of the classification accuracy of the classifiers trained on the extracted confident data. Note that the low data quality in the first run of the inner loop also justifies that a single deep model initialization may lead to sample selection bias.

There are interesting observations of the proposed method that the number of extracted confident examples is close to the number of accurately labeled data in the training set and that the label precision of the extracted confident examples is quite high, i.e., almost all are above 90%. This empirically proves that Me-Momentum is powerful in extracting confident examples. In the next subsection, we will visualize that Me-Momentum is also good at extracting hard confident examples.

## 3.2 VISUALIZE HARD CONFIDENT EXAMPLES

To justify that Me-Momentum is able to extract hard confident examples, we visualize the extracted confident examples by employing t-SNE (Maaten & Hinton, 2008). Specifically, we show how the

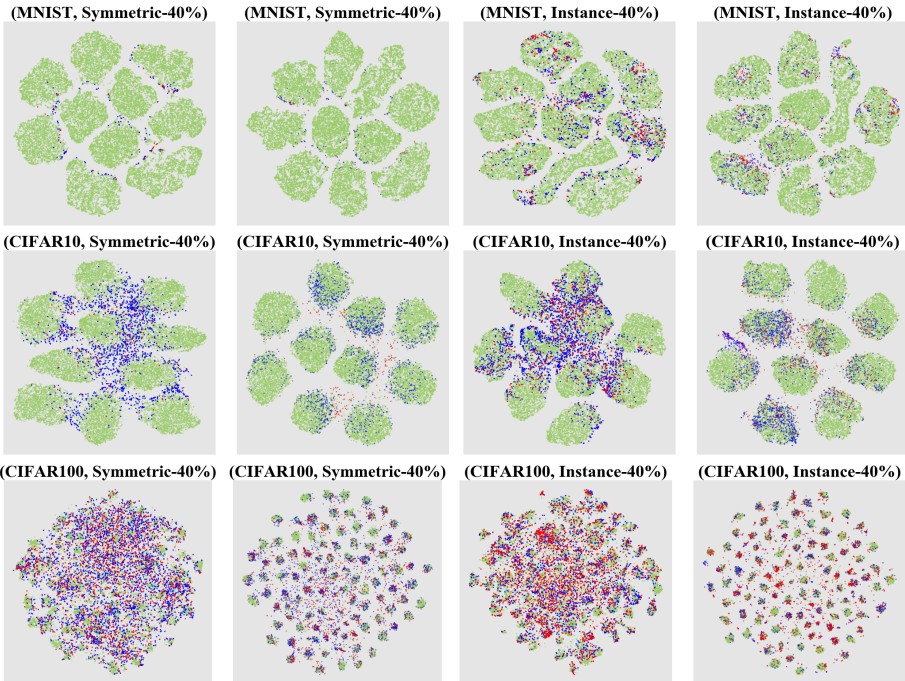

Figure 4: Visualization of the extracted confident examples. The first and third columns are about the confident data extracted in the first run of the inner loop; while the second the fourth columns are about the confident data extracted in the outer loop. Specifically, green dots represent the data selected in the first round. Blue and red dots represent the new extracted data in the middle and the end rounds respectively. Large figures for *CIFAR100* are provided in the supplementary material.

confident examples are progressively extracted in the inner and outer loops. The results are shown in Figure 4, where green, blue, and red dots represent confident examples extracted at the beginning, middle, and end rounds of the loops, respectively. On the datasets of *MNIST* and *CIFAR10*, we can clearly see that the blue and red dots are mostly located at the boundaries of the clusters of green dots. Although the figures of *CIFAR100* are small, we can also clearly see that there are lots of blue and red dots which are outside of the green clusters in the second and fourth figures. This supports and justifies our claim that Me-Momentum is able to extract hard confident examples (those are close to the decision boundary).

Comparing the extracted results of the first run of the inner loop (the first and third columns) with those of the outer loop (the second and fourth columns), we can find that the cluster boundaries in the latter are more clear. This further justifies that why better classification performance can be obtained by re-initialization in the outer loop. Comparing the confident examples extracted on the class-dependent label noise datasets with those on the instance-dependent label noise datasets, we can see that the proposed method is not very sensitive to the type of label noise and can work well on the most general instance-dependent label noise cases.

### 3.3 CLASSIFICATION ACCURACY

**Synthetic data** To evaluate the classification performance of Me-Momentum, we first conduct experiments on *MNIST*, *CIFAR10*, and *CIFAR100* with class-dependent and instance-dependent label noise. Each trial is repeated five times. The results are presented in Tables 1, 2, and 3, respectively. Me-Momentum consistently outperforms the baselines. Specifically, *CIFAR100* is the most challenging one among the three datasets. On *CIFAR100*, Me-Momentum outperforms the baselines by a clear margin across all the settings as shown in Table 3. Note that the performance gain in Me-Momentum is caused by the improvement of the quality of the extracted confident examples.

In the baselines, Co-teaching, Joint Optim, and T-revision are the representative methods that learn robust classifiers by extracting confident examples, refining the noisy labels, and exploiting the noise transition matrix, respectively. Note that Co-teaching keeps updating a constant number of confident

Table 1: Means and standard deviations of classification accuracy on *MNIST*

| Flipping-Rate | Decoupling | MentorNet | Co-teaching | Forward | Joint Optim | DMI | T-revision | Ours |
|---|---|---|---|---|---|---|---|---|
| Sym-20% | 95.39% | 96.57% | 97.22% | 98.22% | 98.58% | 98.92% | 98.91% | **98.94%** |
| | ±0.29% | ±0.18% | ±0.18% | ±0.08% | ±0.15% | ±0.11% | ±0.04% | ±0.13% |
| Sym-40% | 90.77% | 96.16% | 94.64% | 96.71% | 98.12% | 98.63% | 98.42% | **98.66%** |
| | ±0.77% | ±0.49% | ±0.33% | ±0.16% | ±0.06% | ±0.11% | ±0.47% | ±0.07% |
| Inst-20% | 96.94% | 94.66% | 95.37% | 95.89% | 98.10% | 98.75% | 97.12% | **98.96%** |
| | ±0.07% | ±0.35% | ±0.08% | ±0.12% | ±0.14% | ±0.11% | ±0.09% | ±0.06% |
| Inst-40% | 94.98% | 88.51% | 90.06% | 88.95% | 92.00% | 97.58% | 94.89% | **98.11%** |
| | ±0.27% | ±0.36% | ±0.81% | ±2.47% | ±1.39% | ±0.82% | ±0.66% | ±0.35% |

Table 2: Means and standard deviations of classification accuracy on *CIFAR10*

| Flipping-Rate | Decoupling | MentorNet | Co-teaching | Forward | Joint Optim | DMI | T-revision | Ours |
|---|---|---|---|---|---|---|---|---|
| Sym-20% | 79.85% | 80.49% | 87.16% | 85.63% | 89.70% | 88.18% | 89.63% | **91.44%** |
| | ±0.30% | ±0.52% | ±0.11% | ±0.52% | ±0.11% | ±0.36% | ±0.13% | ±0.33% |
| Sym-40% | 69.47% | 77.48% | 83.59% | 74.30% | 87.79% | 83.98% | 86.81% | **88.39%** |
| | ±0.87% | ±3.45% | ±0.28% | ±0.26% | ±0.20% | ±0.48% | ±0.21% | ±0.34% |
| Inst-20% | 77.85% | 79.12% | 86.54% | 85.29% | 89.69% | 89.14% | 90.46% | **90.86%** |
| | ±0.23% | ±0.42% | ±0.11% | ±0.38% | ±0.42% | ±0.36% | ±0.13% | ±0.21% |
| Inst-40% | 59.05% | 70.27% | 80.98% | 74.72% | 82.62% | 84.78% | 85.37% | **86.66%** |
| | ±0.73% | ±1.52% | ±0.39% | ±3.24% | ±0.57% | ±1.97% | ±3.36% | ±0.91% |

Table 3: Means and standard deviations of classification accuracy on *CIFAR100*

| Flipping-Rate | Decoupling | MentorNet | Co-teaching | Forward | Joint Optim | DMI | T-revision | Ours |
|---|---|---|---|---|---|---|---|---|
| Sym-20% | 42.75% | 52.11% | 59.28% | 57.75% | 64.55% | 58.73% | 65.40% | **68.03%** |
| | ±0.49% | ±0.10% | ±0.47% | ±0.37% | ±0.38% | ±0.70% | ±1.07% | ±0.53% |
| Sym-40% | 37.13% | 35.12% | 51.60% | 38.59% | 57.97% | 49.81% | 57.71% | **63.48%** |
| | ±0.91% | ±1.13% | ±0.49% | ±1.62% | ±0.67% | ±1.22% | ±0.84% | ±0.72% |
| Inst-20% | 48.33% | 51.73% | 57.24% | 58.76% | 65.15% | 58.05% | 60.71% | **68.11%** |
| | ±0.35% | ±0.17% | ±0.69% | ±0.66% | ±0.31% | ±0.20% | ±0.73% | ±0.57% |
| Inst-40% | 34.26% | 40.90% | 45.69% | 44.50% | 55.57% | 47.36% | 51.54% | **58.38%** |
| | ±0.59% | ±0.45% | ±0.99% | ±0.72% | ±0.41% | ±0.68% | ±0.91% | ±1.28% |

examples from the mini-batches used in SGD. We therefore do not compare with its extracted confident examples in Section 3.2 as our method extracts confident examples from the whole training data at once. By comparing the classification performance, we can clearly see that the proposed method is much more powerful in extracting confident examples. Note that Joint Optim and T-revision employ all training data to train the classifiers; while our method only employs confident examples and discards the unconfident ones. The results further justify that Me-Momentum is able to extract high-quality confident examples. Note that the performance of Me-Momentum could be further improved by correcting the unconfident data with the idea of Joint Optim.

**Real-world dataset** We compare Me-Momentum with baseline methods on *Clothing1M* in Table 4, where "pre-trained" and "scratch" mean the network was pre-trained by employing *ImageNet* and initialized randomly, respectively; "clean" and "noisy" means the validation data is clean and noisy respectively. First, it is observed that Me-Momentum works well with noisy validation, even surpassing many baselines with clean validation. For a fair comparison, we also use clean data to validate our method, which achieves the highest test accuracy of 75.18%, better than T-revision by 1% and Joint Optim by 2.95%. Note that Forward and T-revision need the 50k clean data for estimating transition matrix, while Me-Momentum does not need any clean data for training. In addition, to show the robustness of Me-Momentum, we conduct experiments with ResNet-50 from scratch, which achieves the second best accuracy.

Table 4: Classification accuracy on *Clothing1M*.

| Method | Validation | Accuracy |
|---|---|---|
| Cross Entropy | Clean | 69.54% |
| Decoupling | Clean | 53.98% |
| MentorNet | Clean | 56.77% |
| Co-teaching | Clean | 58.68% |
| Forward | Clean | 69.84% |
| Joint Optim | Clean | 72.23% |
| DMI | Clean | 72.46% |
| T-revision | Clean | 74.18% |
| Ours (pre-trained) | Noisy | 73.13% |
| Ours (scratch) | Clean | 74.75% |
| Ours (pre-trained) | Clean | **75.18%** |

## 4 CONCLUSION

In this paper, we propose a method called Me-Momentum that is able to extract hard confident examples from noisily labeled data by exploiting the memorization effect of deep neural networks. In a high level, it fulfils a positive cycle that better confident examples will result in a better classifier and that a better classifier will identify better confident examples. We have empirically verified its effectiveness by analyzing the statistics of the extracted examples, visualizing the hard confident examples, and comparing its classification performance with state-of-the-art baselines. In future, we can extend our work by utilizing and exploiting the unconfident examples.

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

# A    EARLY STOPPING

We discuss the early stopping trick used in Step 1 of Algorithm 1. We repeat explaining the trick here again. Specifically, we introduce a definition of *high-peak*. A noisy validation accuracy at the $i$-th epoch is called an $i$-th high-peak if it achieves the highest accuracy in the epoch range $\{1, \ldots, i\}$. Assume the $i$-th and $j$-th high-peaks occur next to each other, having noisy validation accuracies of $a$ and $b$, respectively. The training early stops if $(b - a)/(j - i) \leq \tau$, where $\tau$ is a hyper-parameter. In the experiments, we set $\tau = 0.1$, which empirically works well across all datasets. In Figure 5, we compare the difference between the early stopping method and the traditional validation method.

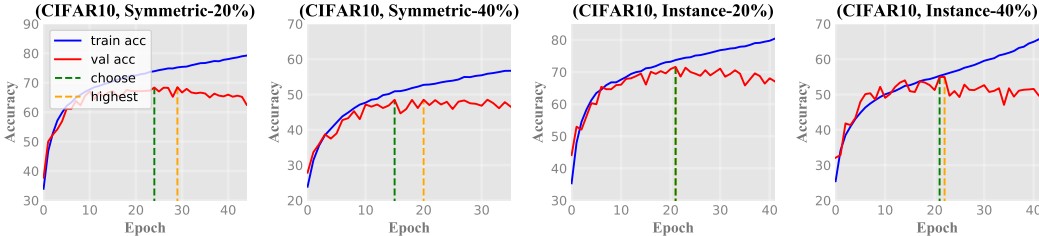

Figure 5: Comparing the difference between the early stopping method in Step 1 and the traditional validation method where the classifier with the highest validation accuracy during the whole training procedure will be output. The green dash line indicates the epoch at which early stopping happens; while the orange dash line indicates the epoch at which the highest validation accuracy is achieved during the whole training procedure. In the third plot, the two dash lines are identical to each other.

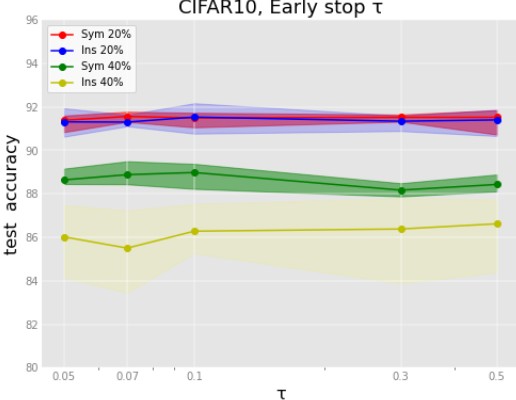

Figure 6: Illustrative of extracting hard confident examples.

We also study the sensitivity of the hyper-parameter. Specifically, we study its sensitivity on *CI-FAR10* by setting $\tau$ to be the values in the range $\{0.05, 0.07, 0.1, 0.3, 0.5\}$. Other settings are the same as those in this paper. The results are presented in Figure 6. We can see that the classification performance of Me-Momentum is robust and not sensitive to the change of the value of $\tau$.

# B    GENERATING LABEL NOISE

**Transition matrix**   The *transition matrix* $T(\boldsymbol{x})$ was proposed to explicitly model the generation process of label noise, where $T_{ij}(\boldsymbol{x}) = \Pr(\bar{Y} = j | Y = i, X = \boldsymbol{x})$, $\Pr(A)$ denotes as the probability of the event $A$, $X$ as the random variable for the instance, $\bar{Y}$ as the noisy label, and $Y$ as the latent clean label. In high level, the $ij$-th entry of the transition matrix denote the probability that the instance will flip from the clean class $j$ to the noisy class $i$.

**Symmetric class-dependent label noise**   The label noise is generated according to symmetric class-dependent and instance-independent noise transition matrices. If the flip rate is $\alpha$, the diagonal entries of a symmetric transition matrix are $1 - \alpha$ and the off-diagonal entries are $\alpha/(c - 1)$.

**Instance-dependent label noise** We generate the instance-dependent label noise according to the following Algorithm 2 Xia et al. (2020). More details about this algorithm can be found in Xia et al. (2020).

---

**Algorithm 2** Instance-dependent Label Noise Generation

---

**Input**: Clean samples $\{(\boldsymbol{x}_i, y_i)\}_{i=1}^n$; Noise rate $\tau$.
1: Sample instance flip rates $q \in \mathbb{R}^n$ from the truncated normal distribution $\tau \mathcal{N}(\tau, 0.1^2, [0, 1])$;
2: Independently sample $w_1, w_2, \ldots, w_c$ from the standard normal distribution $\mathcal{N}(0, 1^2)$;
3: For $i = 1, 2, \ldots, n$ do
4:    $p = \boldsymbol{x}_i \times w_{y_i}$;                                 // generate instance-dependent flip rates
5:    $p_{y_i} = -\infty$;            // control the diagonal entry of the instance-dependent transition matrix
6:    $p = q_i \times softmax(p)$;       // make the sum of the off-diagonal entries of the $y_i$-th row to be $q_i$
7:    $p_{y_i} = 1 - q_i$;                                  // set the diagonal entry to be $1 - q_i$
8:    Randomly choose a label from the label space according to possibilities $p$ as noisy label $\bar{y}_i$;
9: End for.
**Output**: Noisy samples $\{(\boldsymbol{x}_i, \bar{y}_i)\}_{i=1}^n$

---

## C   COMPARE RESULTS WITH SELF

The performance of our re-implemented SELF (Tarvainen & Valpola, 2017) is not as good as that in the original paper. For a fair comparison, we change our backbone consistently with SELF and compare with the results from the original paper directly (without Mean Teacher (Tarvainen & Valpola, 2017)). Specifically, the network is changed to ResNet26 with Shake-shake regularization (Gastaldi, 2017) and the learning rate is changed to 0.05 and weight decay to 2e-4.

Table 5 shows that Me-Momentum outperforms SELF by a large margin in CIFAR10 and CIFAR100 with Symmetric 40% and Symmetric 60% noise. The gap between the performance of Me-Momentum and SELF becomes larger in Symmetric 60% in both datasets compared with Symmetric 40% because hard confident examples play a more important role in more noisy datasets. Specifically, both of the methods are based on the same backbone with Cross-Entropy loss, so the improvement of Me-Momentum can only be as a result of the quality of extracted confident examples. Therefore, Me-Momentum is able to extract better hard confident examples than SELF.

Table 5: Means and standard deviations of classification accuracy compared with SELF

|  | CIFAR10 Sym-40 | CIFAR10 Sym-60 | CIFAR100 Sym-40 | CIFAR100 Sym-60 |
|---|---|---|---|---|
| SELF | 87.35% | 75.47% | 61.40% | 50.60% |
| Ours | **92.31%** | **87.88%** | **68.25%** | **59.51%** |

Furthermore, results in Table 5 shows that the performance of Me-Momentum can be improved by changing a better backbone. However, to show the effectiveness of the proposed method and avoid complexity, we simply choose in the paper the standard CNN network.

## D   EXPERIMENTS COMPLEMENTARY TO SECTION 3.1

In Section 3.1, we discuss the statistics of the extracted confident examples. However, due the space limit, in the paper, we only provide parts of the empirical results. In this supplementary material, we provide the statistics on all the employed datasets and settings.

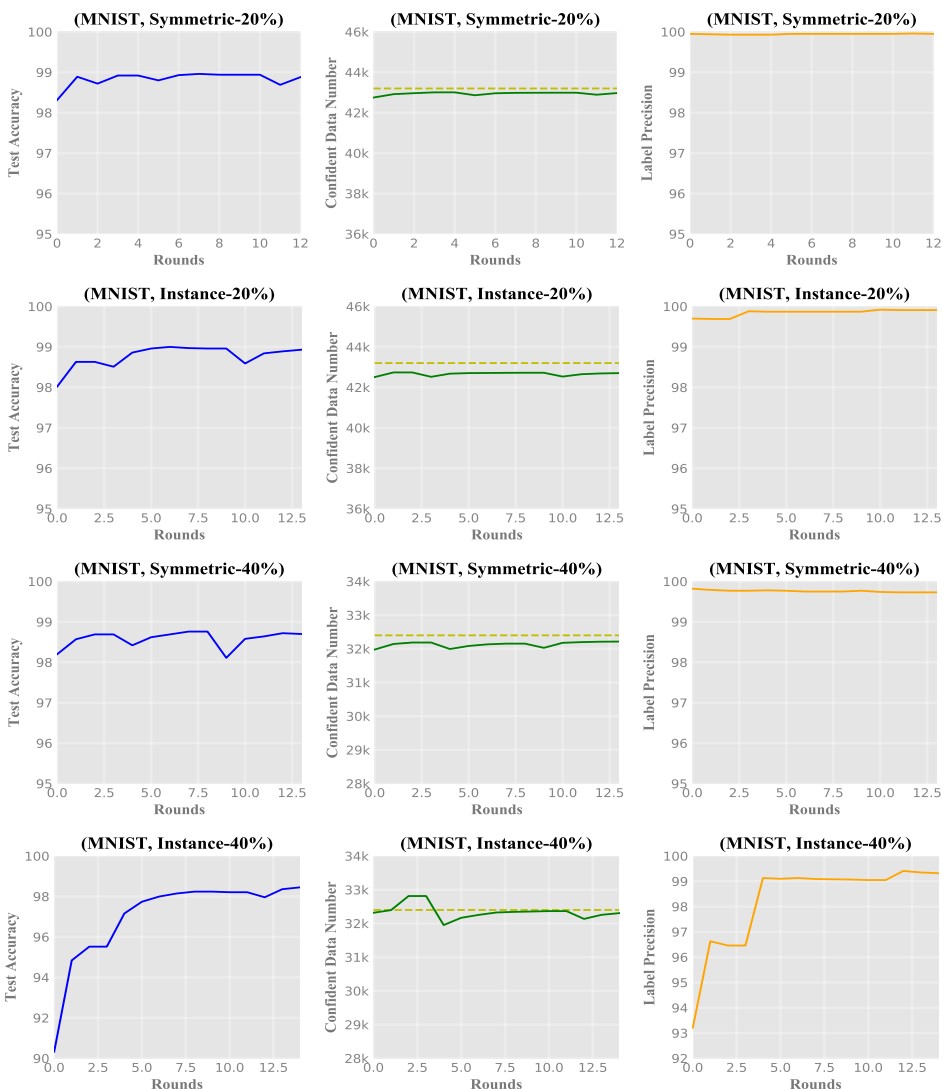

Figure 7: Statistics of the extracted confident examples on *MNIST* by Me-Momentum. We call one update of the classifier and the extracted confident examples as one round. We illustrate how the label precision of the extracted confident examples, the number of the extracted confident examples, and the classification accuracy of the classifier trained by using the extracted confident examples change during the training of Me-Momentum. The dash lines in the middle column indicate the number of clean labels in the noisy training data.

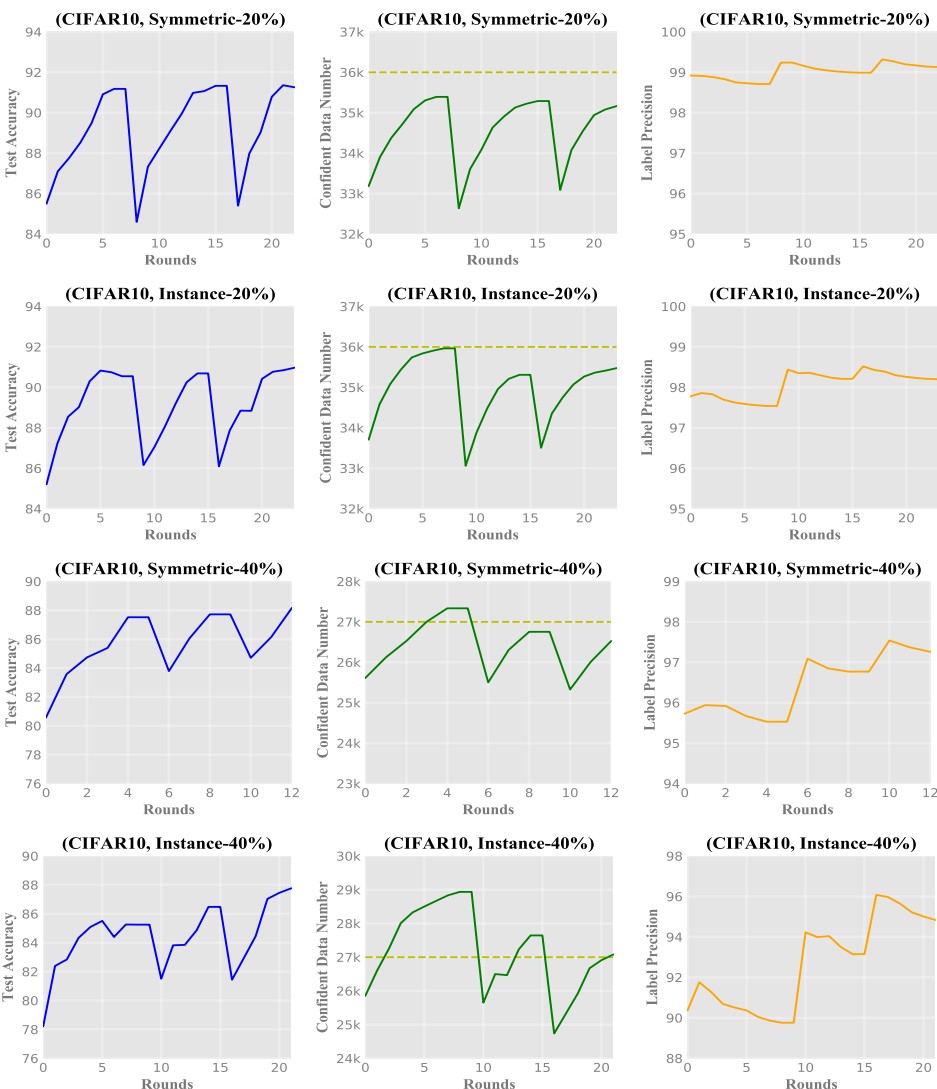

Figure 8: Statistics of the extracted confident examples on *CIFAR10* by Me-Momentum. We call one update of the classifier and the extracted confident examples as one round. We illustrate how the label precision of the extracted confident examples, the number of the extracted confident examples, and the classification accuracy of the classifier trained by using the extracted confident examples change during the training of Me-Momentum. We have three distinct peaks in these figures because we have set $N_{\text{outer}} = 3$ and the classifiers are re-initialized in the outer loop. The dash lines in the middle column indicate the number of clean labels in the noisy training data.

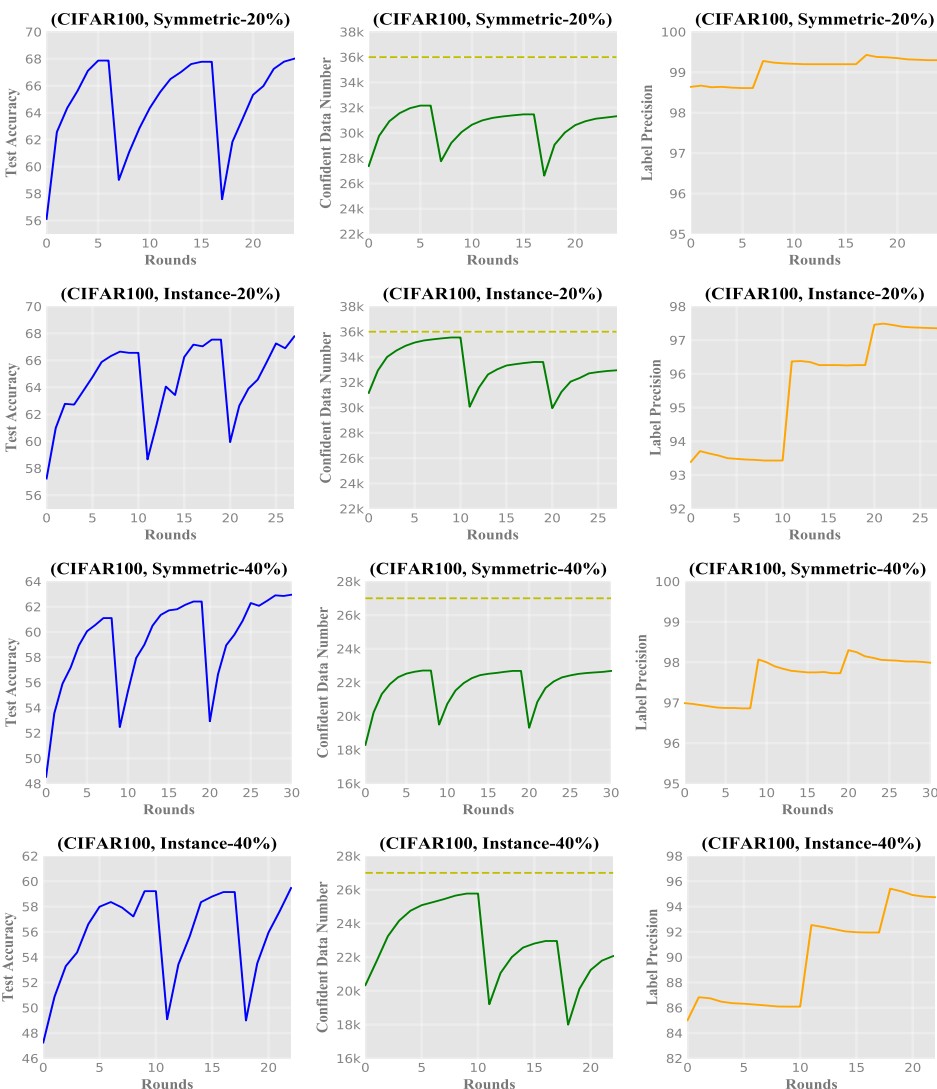

Figure 9: Statistics of the extracted confident examples on *CIFAR100* by Me-Momentum. We call one update of the classifier and the extracted confident examples as one round. We illustrate how the label precision of the extracted confident examples, the number of the extracted confident examples, and the classification accuracy of the classifier trained by using the extracted confident examples change during the training of Me-Momentum. We have three distinct peaks in these figures because we have set $N_{\text{outer}} = 3$ and the classifiers are re-initialized in the outer loop. The dash lines in the middle column indicate the number of clean labels in the noisy training data.

## E    EXPERIMENTS COMPLEMENTARY TO SECTION 3.2

In Section 3.2, we have visualize the extracted confident examples by using t-SNE. It verifies that the proposed Me-Momentum is effective to extract hard confident examples. However, due the space limit, in the paper, we only provide parts of the empirical results. In this supplementary material, we provide the visualization on all the employed datasets and settings.

Specifically, we show how the confident examples are progressively extracted in the inner and outer loops. In the figures, green, blue, and red dots represent confident examples extracted at the beginning, middle, and end rounds of the loops, respectively.

On the datasets of *MNIST* and *CIFAR10*, we can clearly see that the blue and red dots are mostly located at the boundaries of the clusters of green dots. On *CIFAR100*, we can also clearly see that

there are lots of blue and red dots which are outside of the green clusters in the second and fourth figures. This supports and justifies our claim that Me-Momentum is able to extract hard confident examples (those are close to the decision boundary).

**The figures are presented in the next and the following pages.**

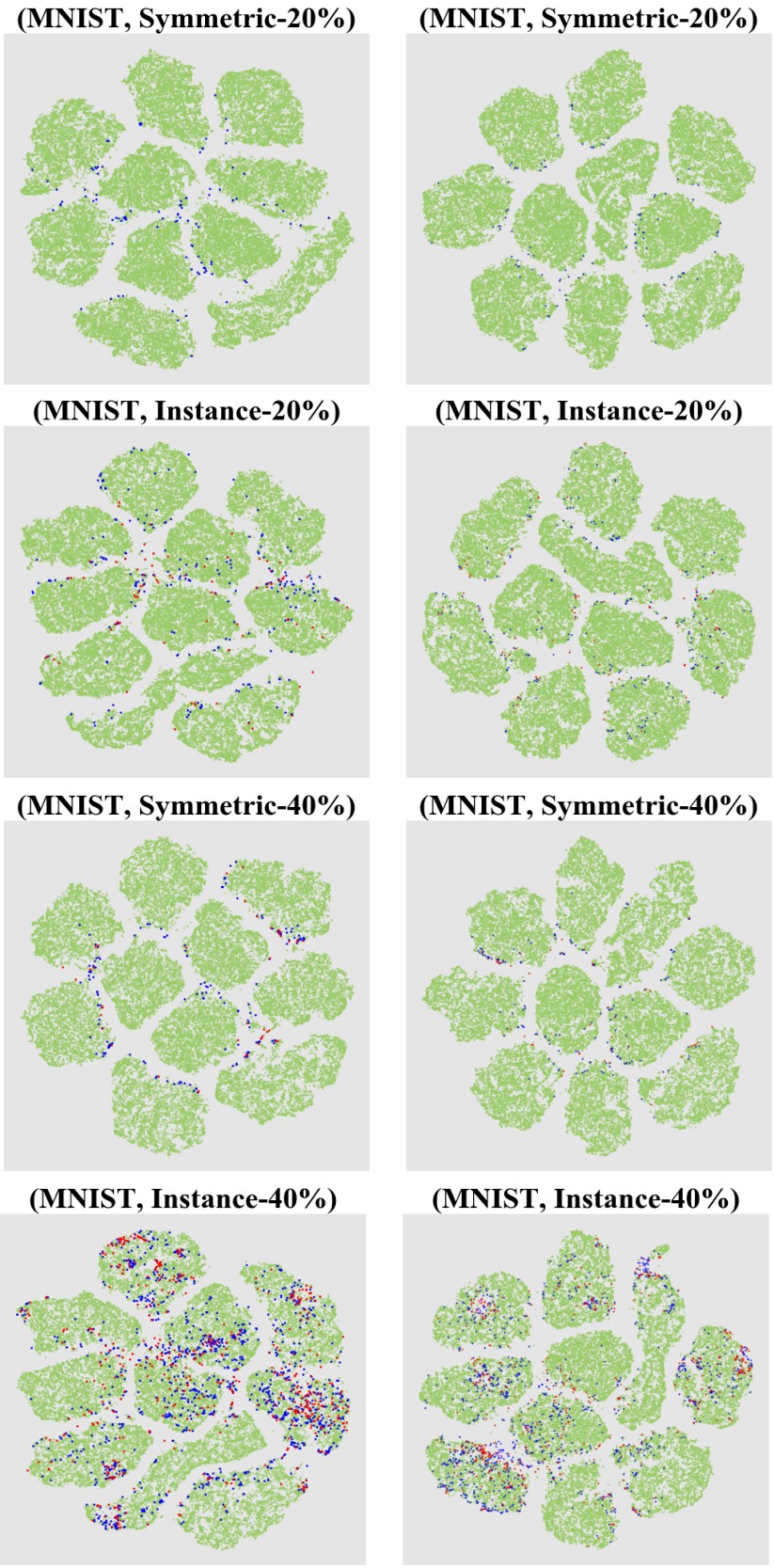

Figure 10: Visualization of the extracted confident examples on *MNIST*. The first column is about the confident data extracted in the first run of the inner loop; while the second column is about the confident data extracted in the outer loop.

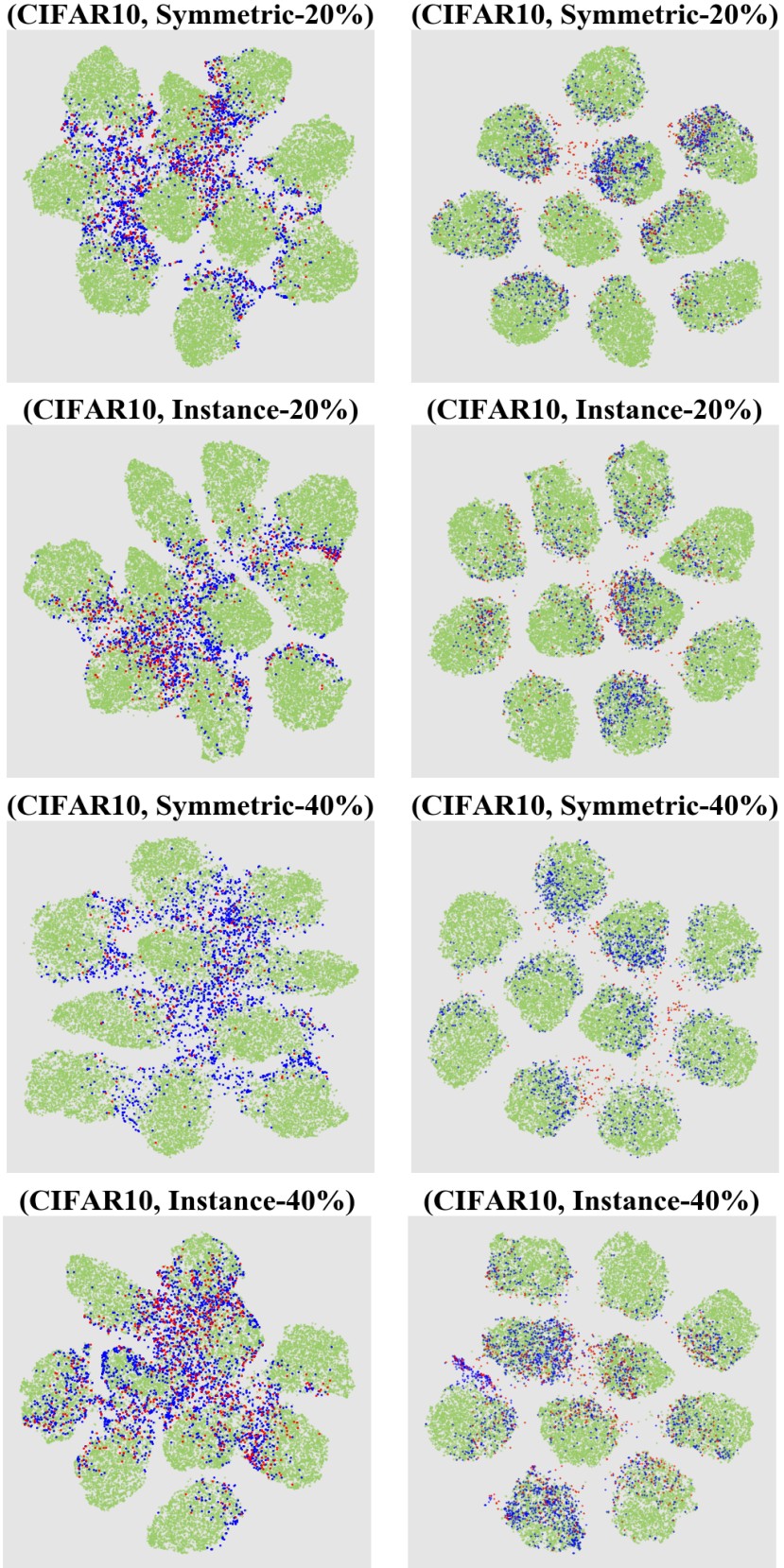

Figure 11: Visualization of the extracted confident examples on *CIFAR10*. The first column is about the confident data extracted in the first run of the inner loop; while the second column is about the confident data extracted in the outer loop.

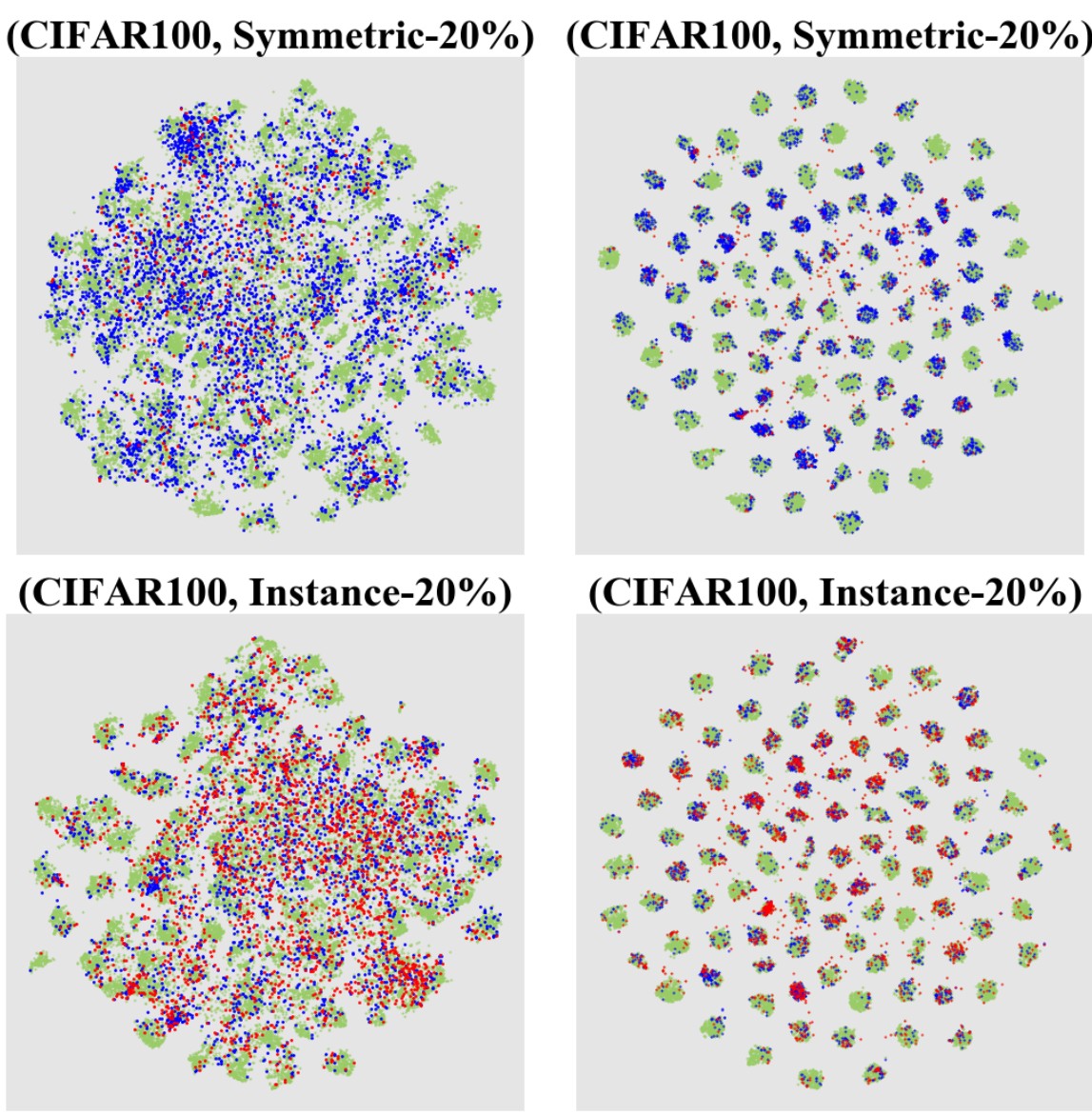

Figure 12: Visualization of the extracted confident examples on *CIFAR100*. The first column is about the confident data extracted in the first run of the inner loop; while the second column is about the confident data extracted in the outer loop.

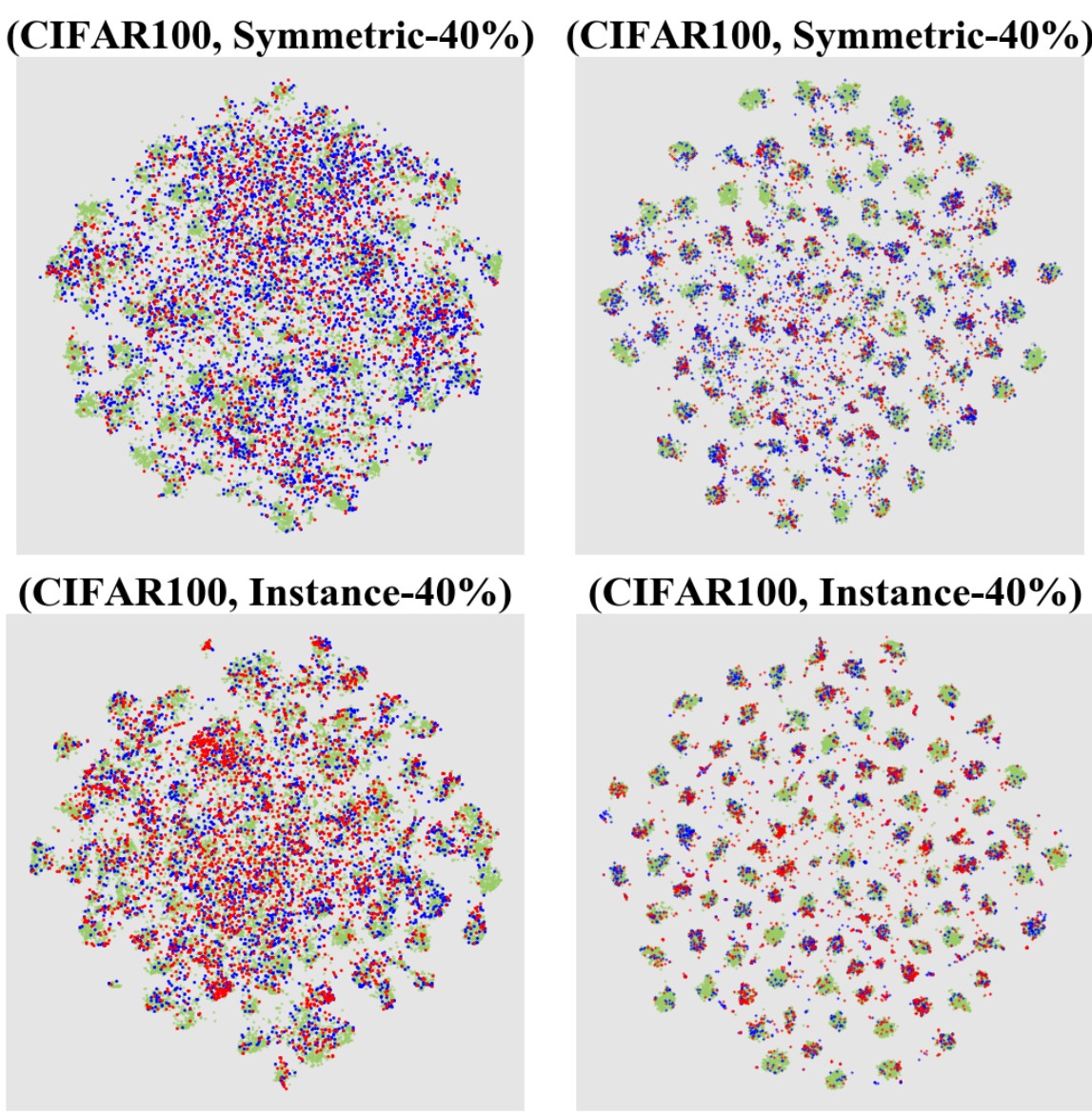

Figure 13: Visualization of the extracted confident examples on *CIFAR100*. The first column is about the confident data extracted in the first run of the inner loop; while the second column is about the confident data extracted in the outer loop.

