# OpenReview forum: "ME-MOMENTUM:  EXTRACTING HARD CONFIDENT EXAMPLES FROM NOISILY LABELED DATA"
_ICLR.cc/2021/Conference — Reject_

### Official Review · AnonReviewer1 · 2020-10-23
**Potentially interesting work, but technical contribution is limited, and the method is not theoretically solid.**

**Rating:** 4
**Confidence:** 4

**Review:**

*quality*
The organization of this paper is barely satisfactory. The technique proposed to extract hard confident examples in this paper is not convincing, even though the experimental results seem promising.

*clarity*
It is not difficult to understand the proposed method, however, the Figures in this paper are somewhat confusing to readers.

*originality*
In this paper, the authors focus on extracting hard confident examples from the noisy training data for learning with noisy labels. There is no method to extract hard confident examples before, the idea is novel. However, the technical contribution is limited.

*significance*
The idea of extracting hard confident examples is good, however, I do not think this paper solves this issue properly. How to extract hard confident examples correctly still remains a challenging problem.

*pros and cons*
Pros:
(1). Authors provide sufficient experiments which evaluate the methods on multiple datasets.
(2). The proposed method achieves better performance than state-of-the-art methods.

Cons:
(1). As for the question “How to validate the learned classifiers in Steps 3 and 5 without a clean validation set?”, the authors claim that the noisy validation set could be used as a surrogate to validate the classifiers if no clean validation set is available. I do not think it is reasonable here.
(2). The confident examples can be extracted based on the memorization effect, it seems reasonable. But why hard confident examples can be extracted? The authors explained as previously extracted confident examples will help identify hard confident examples, it is so empirical and maybe another explanation is needed here. Moreover, in the corresponding paragraph, there might be some error with “Figure 6” (should be “Figure 2”?).
(3). The intuition “better confident examples will result in a better classifier and a better classifier will identify better confident examples” have been mentioned many times, I wonder will the method proposed in this paper stuck in the local optimum when the two loops in Algorithm 1 execute.
(4). The proposed method performs better than the state-of-the-art with 20% and 40% noisy labels. It would be better if the experiments with lager noise rate (e.g., >=50%) are also conducted.
(5). Some minor issues, for example, “extract” should be “extracted” in the step 2 of Algorithm 1; “$\times$” and “*” are abused in Section 3; and the resolution of some Figures in this paper is not satisfactory.

Generally, I feel that the method proposed in this paper is somehow too empirical, and more theoretical study is needed.

---

> ### Author Response · Authors · 2020-11-17
> **To AnonReviewer1**
>
> Q1: About the noisy validation set.
> A1: Our work follows the widely used practice in the literature of learning from noisy labels, i.e., using a noisy validation set for early stopping [r2][r3][r4][r5]. It is empirically reported to work well. Note that the correct labels are dominating in each noisy class and that label noise is random, the accuracy on the noisy validation set, and the accuracy on the clean test data set are positively correlated. The noisy validation set therefore can be employed. We agree with the reviewer that a more in-depth theoretical analysis of this aspect is worth further learning. However, we respectively cannot agree with the reviewer that using a noisy validation set is not acceptable.
>
> [r2] Zhilu Zhang and Mert Sabuncu. Generalized cross-entropy loss for training deep neural networks with noisy labels. NeurIPS, 2018.
> [r3] Giorgio Patrini, Alessandro Rozza, Aditya Krishna Menon, Richard Nock, and Lizhen Qu. Making deep neural networks robust to label noise: A loss correction approach. CVPR, 2017.
> [r4] Xiaobo Xia, Tongliang Liu, Nannan Wang, Bo Han, Chen Gong, Gang Niu, and Masashi Sugiyama. Are anchor points really indispensable in label-noise learning? NeurIPS, 2019.
> [r5] Xiaobo Xia, Tongliang Liu, Bo Han, Nannan Wang, Mingming Gong, Haifeng Liu, Gang Niu, Dacheng Tao, and Masashi Sugiyama. Part-dependent label noise: Towards instance-dependent label noise. NeurIPS 2020.
>
> Q2: Why hard confident example can be extracted.
> A2: Note that hard confident examples are entangled with incorrectly labeled data. If we just use the traditional memorization-based trick, e.g., the small-loss trick, we cannot distinguish the hard confident examples and the incorrectly labeled data. However, with the “Momentum of memorization”, we can do so. Specifically, by using the traditional memorization-based trick, we could extract easy confident examples and learn classifiers based on them. The hard confident examples is more connected to the easy confident examples than the incorrectly labeled examples because the former two share some common features. By using the classifiers learned by easy confident examples, we can distinguish the hard confident examples from the incorrectly labeled data and thus extract hard confident examples. This philosophy is called the memorization of the momentum. We will make this clear in the paper.
>
> Q3:  Stuck in the local optimum
> A3: Thanks for the nice concern. By simply implementing the intuition, it is likely that we will stick in the bad local optimum because we have the memorization of noisy labels and the inferiority of sample-selection bias. However, this could be effectively addressed by reinitializing the classifier at the beginning of each outer loop. Specifically, with the noisy dataset, especially without a clean validation set, the classifier may be affected by label noise and sticks into a bad local optimum due to the memorization of neural networks. Reinitializing neural networks provide an opportunity to relearn the knowledge in a lower noisy dataset compared with the last outer loop. This helps the classifier escape the bad local optimum. Evidence is that the standard deviations of classification accuracy in three artificial datasets are relatively low compared with other baselines.
>
> Q4: Higher noise rate
> A4: Note that we only make use of confident examples and discard the non-confident examples. If the noise rate is too high, e.g., noise rate 80%, the number of extracted examples may become too small, which means there are not enough examples for good training. This could be addressed by making use of the non-confident examples by using the semi-supervised learning method, e.g., DivideMix [r1]. However, our aim is to verify the effectiveness of the method to extract high-quality confident examples, not to boost the classification performance.
>
> We think even with the currently used noise rates, the empirical results of the proposed method are convincing, showing that we indeed extracted high-quality (and hard) confident examples. Here are results that comparing with baselines on Symmetric 50%
>
> Dataset       JointOptim   DMI      T-revision  Me-Momentum
> CIFAR10 Sym-50   85.00±0.17    78.28±0.48    83.49±0.42  86.40±0.34
> CIFAR100 Sym-50    50.22±0.41    44.25±1.14    49.28±0.43  58.06±0.59
>
> Note that we compared the proposed method with SELF on Symmetric 60% in appendix C (backbone consistently with SELF). Specifically,
>
> Dataset       SELF     Me-Momentum
> CIFAR10 Sym-60   75.47%    87.88%
> CIFAR100 Sym-60    50.60%    59.51%
>
> The results show that Me-Momentum largely outperforms SELF. Note that the backbone and loss function (cross-entropy loss function) are the same. The outperformance clearly shows that Me-Momentum can better extract high-quality confident examples significantly.
>
> [r1] Li, Junnan, Richard Socher, and Steven CH Hoi. "DivideMix: Learning with Noisy Labels as Semi-supervised Learning." In ICLR. 2019.

---

### Official Review · AnonReviewer2 · 2020-10-26
**Review 2 for "ME-MOMENTUM: EXTRACTING HARD CONFIDENT EXAMPLES FROM NOISILY LABELED DATA"**

**Rating:** 7
**Confidence:** 4

**Review:**

This paper propose a novel and effective method called Me-Momentum to cope with noisy labels. The algorithm borrows the idea of momentum from physics and tries to identify hard examples. The authors alternately update the hard examples and improve the classifier to achieve the robustness to noisy labels. Experiments and comparisons with recent state-of-art methods are provided to verify the effectiveness of Me-Momentum.

Pros:

1. Different from the existing methods, which aim to identify simple clean examples, this paper analyzes the importance of hard examples and provide a way to identify them. The method is interesting and effective. It gives a new perspective and makes clear contributions for learning with noisy labels.

2. The design of inner loop and outer loop is interesting and insightful, and is proved to be very effective.

3. The experimental results are promising. The proposed method achieves great performance (75.18%) on Clothing1M. In addition, compared with SOTA SELF,  Me-Momentum outperforms it by a large margin. The authors also provide an ablation study to analyze the sensitivity of the hyperparameter $tau$. Thus, the significance of the proposed method with respect to experimental results may be high in the community.

4. The paper is well written. The description of the technical details is very clear. It is easy to reproduce this method.

Specific comments and questions:

1. Insufficient analysis of visualization results and analysis of hyperparameter sensitivity. The authors should add them to improve this paper. In addition, the baselines are reimplemented with default parameters? I hope the author can emphasize or add some descriptions of the implementation details of the comparison methods.

2. Adding more baselines will be better. Some recent methods achieve great classification performance for learning with noisy labels, such as [1]. The authors can compare the proposed method with them to make the results more convincing.

3. The proposed method uses a noisy validation set to choose classifiers, and then identify confident examples with robust classifier. The authors should explain why a noisy validation dataset can be used to choose classifiers which perform well on clean datasets. This choice may not be accurate?

4. There are some grammatical errors and typos. The author should proofread this paper. For example, “The results are presented in Figure XX”, the authors miss the figure number. Please check it carefully.

[1] NLNL: Negative Learning for Noisy Labels, Youngdong Kim et al, ICCV2019.

---

> ### Author Response · Authors · 2020-11-17
> **To  AnonReviewer2**
>
> Thanks for the positive support!
>
> Q1: Visualization analyses and hyperparameter sensitivity
> A1: As pointed out by reviewer 3 that t-SNE is known to preserve local structure better than global structure, making it a poor way to visualize how close confident examples are to decision boundaries. However, the visualization at least shows that we could extract confident examples that are far away from the cluster centroids. Those could be interpreted as hard confident examples as well. The current figures clearly show that we gradually extract confident examples on the boundaries of the clusters. The label precision in Figure 3 shows that the vast majority of the extracted confident examples are of correct labels. The high classification accuracy also verifies that the extracted confident examples are of high quality.
>
> Figure 6 shows that the proposed method is not sensitive to the hyperparameter tao, which is proposed for robust early stopping. We may not need to set specific values for N_outer and N_inner as we can stop the algorithm when there no improvement in the next inner or outer loop by looking at the validation set. However, this may be time-consuming. We set N_outer and N_inner to be specific values to save time.
>
> From Figure 3, we can see how the number of confident examples, confident label precision, and the test accuracy varies by increasing the number of the round of inner and outer loops in a limited range. We can see that the curves are quite smooth, which shows that the proposed method is not very sensitive to the hyperparameters.
>
>
> Q2: Adding more baselines
> A2: Our aim is to verify the effectiveness of the method to extract high-quality confident examples, not to boost the classification performance. The not-so-confident examples could be exploited, e.g., by the semi-supervised learning methods, to further improve the performance as did in DivideMix [r1].
>
> [r1] Li, Junnan, Richard Socher, and Steven CH Hoi. "DivideMix: Learning with Noisy Labels as Semi-supervised Learning." In ICLR. 2019.
>
>
> Q3: Why noisy validation set
> Q3: Our work follows the widely used practice in the literature of learning from noisy labels, i.e., using a noisy validation set for early stopping [r2][r3][r4][r5]. It is empirically reported to work well. Note that the correct labels are dominating in each noisy class and that label noise is random, the accuracy on the noisy validation set, and the accuracy on the clean test data set are positively correlated. The noisy validation set therefore can be employed. A more in-depth theoretical analysis of this aspect is worth further learning.
>
> [r2] Zhilu Zhang and Mert Sabuncu. Generalized cross-entropy loss for training deep neural networks with noisy labels. NeurIPS, 2018.
> [r3] Giorgio Patrini, Alessandro Rozza, Aditya Krishna Menon, Richard Nock, and Lizhen Qu. Making deep neural networks robust to label noise: A loss correction approach. CVPR, 2017.
> [r4] Xiaobo Xia, Tongliang Liu, Nannan Wang, Bo Han, Chen Gong, Gang Niu, and Masashi Sugiyama. Are anchor points really indispensable in label-noise learning? NeurIPS, 2019.
> [r5] Xiaobo Xia, Tongliang Liu, Bo Han, Nannan Wang, Mingming Gong, Haifeng Liu, Gang Niu, Dacheng Tao, and Masashi Sugiyama. Part-dependent label noise: Towards instance-dependent label noise. NeurIPS 2020.

---

### Official Review · AnonReviewer3 · 2020-10-28
**Simple idea, but many questions about novelty, presentation, and methodology**

**Rating:** 4
**Confidence:** 3

**Review:**

This paper proposes momentum of memorization as a way to distinguish hard examples needed for efficient learning from noisy examples which decrease classification accuracy. The method finds confident, hard examples and updates them dynamically during model training. This is done by iteratively selecting examples with labels that agree with model predictions and then training on only the confident data. Results show improved accuracy on standard image classification datasets with both synthetic and real world label noise.


Novelty
- In addition to the methods cited in "Relation to existing work," the authors should cite and discuss self-training methods which have become quite popular over the past decade [1]
- The paper also describes Algorithm 1 as adding an outer loop to the recent SELF algorithm [2], which considerably limits its novelty.

Correctness/Experiments:
- Results show consistent improvement over baselines
- A more thorough evaluation would include ablation on the parameters $N_{outer}$ and $N_{inner}$, as well as the modified early stopping "trick"
- Figure 4 shows qualitative t-SNE visualizations of identified confident examples. qualitatively, these look well separated. However, t-SNE is known to preserve local structure better than global structure, making it a poor way to visualize how close confident examples are to decision boundaries. It's also impossible to tell whether any of the dots (in particular blue and red) are incorrectly labeled wrt ground truth class labels (i.e. "noisy" vs "hard")
- Many claims are made without theoretical or empirical justification. For example, showing how training/validation accuracy varied with $N_{outer}$ and $N_{inner}$ would support the claim that Me-Momentum is leveraging the memorization effect.

Clarity
- Example figures and problem formulation are quite nice
- I believe this naming may cause confusion with the momentum update widely used in optimization. Additionally, I am not sure how this paper fits the notion of momentum more than any number of other training procedures. For example, increasing the learning rate at each epoch would fit the colloquial/physics meaning of "traveling through hypothesis space" just as well as this approach which uses repeated fine tuning.
- Algorithm 1 and Section 2 would benefit from additional notation in addition to the pseudocode and text explanation
- Unclear what the reader should take away from Figure 4 and 10-13 (see above)
- typos: "an surrogate" should be "a surrogate", and "clothing1M" should be "Clothing1M"

Pro
- Nice illustrations and problem description
- Results perform well against baselines

Cons
- Limited novelty compared to SELF and self-training methods more generally
- Concerns about methodology and clarity, especially wrt visualizations
- Empirical methodology/analysis should be improved

Questions:
- How does the running time compare to normal neural network training, and to other baseline methods?
- Do Tables 1-3 show a similar improvement to Table 4 when switching to clean validation set?
- How sensitive is Me-Momentum to hyperparameters $N_{outer}$ and $N_{inner}$?
- Is the recall of confident examples wrt accurately labeled data as high as the precision? Will identifying additional confident samples lead to even higher accuracy?


[1] Huang and Harper. Self-Training PCFG Grammars with Latent Annotations
Across Languages, EMNLP 2009 https://www.aclweb.org/anthology/D09-1087.pdf

[2] Nguyen et al. SELF: learning to filter noisy labels with self-ensembling, ICLR 2020. https://arxiv.org/abs/1910.01842

EDIT: The author response addressed some of my concerns. In particular, it confirms that the experimental results are impressive compared to many baselines. However, I would appreciate the distinction between easy and hard confident examples much more if the authors went beyond illustrative figures and defined this concept more precisely. Without a precise definition, it's difficult to verify the paper's claims about why the method performs well. Based on t-SNE visualizations, the author response offers an alternate definition of "far away from the cluster centroids." The submission would be much stronger if it developed this idea further and analyzed it quantitatively.

Next, the authors suggest that methods cannot distinguish hard confident examples from mislabeled examples using the "small loss trick" alone, and that their "momentum trick" is necessary. However, they do not present a principled argument or strong evidence to support the claim. In fact, some recent methods do show a separation between these types of training data using measurable quantities, see Figure 1 of [3] and Figures 1-2 of [4].

Finally, the authors claim that reinitialization helps escape bad local optima. However, I do not see how low standard deviation supports this claim.

[3] Pleiss et al. Identifying Mislabeled Data using the Area Under the Margin Ranking, Neurips 2020. https://arxiv.org/abs/2001.10528

[4] Swayamdipta et al. Dataset Cartography: Mapping and Diagnosing Datasets with Training Dynamics, EMNLP 2020. https://arxiv.org/abs/2009.10795]

---

> ### Author Response · Authors · 2020-11-17
> **To AnonReviewer3 (2)**
>
> Regarding the clarity.
> Q5: Regarding the term “Momentum”
> A5: We agree with the reviewer that the term “Momentum” may cause confusion for readers. The examples raised by the reviewer about “Momentum” are also more from the optimization perspective. We will make it clear that the “Momentum” in this paper refers to the “Momentum of memorization”.  Note that the hard confident examples are entangled with incorrectly labeled data. If we just use the traditional memorization-based trick, e.g., the small-loss trick, we cannot distinguish the hard confident examples and the incorrectly labeled data. However, with the “Momentum of memorization”, we can do so. Specifically, by using the traditional memorization-based trick, we could extract easy confident examples and learn classifiers based on them. The hard confident examples are more connected to the easy confident examples compared with the incorrectly labeled examples. By using the classifiers learned by easy confident examples, we can distinguish the hard confident examples from the incorrectly labeled data and thus extract hard confident examples. This philosophy is called the memorization of the momentum. We will make this clear in the paper.
>
>
> Q6: Regarding t-SNE
> A6: We agree with the reviewer that t-SNE is known to preserve local structure better than global structure, making it a poor way to visualize how close confident examples are to decision boundaries. However, the visualization at least shows that we could extract confident examples that are far away from the cluster centroids. Those could be interpreted as hard confident examples as well.
>
> If we further divide the confident examples with correct and correct labels. The figures may look very complicated. The current figures clearly show that we gradually extract confident examples on the boundaries of the clusters. The label precision in Figure 3 shows that the vast majority of the extracted confident examples are of correct labels. The high classification accuracy also verifies that the extracted confident examples are of high quality.
>
> Regarding the novelty.
> Q7: the self-training method.
> A7: Thanks for pointing out the related work. Note that the baselines we compared do not include the state-of-the-art methods which make use of the non-confident examples by using a semi-supervised learning method, e.g., DivideMix [r1]. Our aim is to verify the effectiveness of the method to extract high-quality confident examples, not to boost the classification performance. The self-training method is a powerful semi-supervised learning method. We will discuss it in the related work but will not include it as a baseline. We note that combining the proposed method with self-training is an interesting future work.
>
> [r1] Li, Junnan, Richard Socher, and Steven CH Hoi. "DivideMix: Learning with Noisy Labels as Semi-supervised Learning." In ICLR. 2019.
>
> Q8: Similarity to SELF.
> A8: The two methods are quite different because (1) we have different aims and (2) the methodologies are different.
>
> Especially, SELF is focusing on removing non-confident labels while we are focusing on how to extract high-quality confident examples. As SELF will make use of non-confident examples by using semi-supervised methods, they do not care very much about the hard confident examples. However, identifying the hard confident examples is the bottleneck for our proposed method as we only make use of the confident examples.
>
> Because of the different aims, the methods designed are quite different. Although we use a similar method to identify confident examples, Me-momentum is specifically designed for extracting hard confident examples. A specific early stopping method and re-initialization have been designed for the outer loop to avoid the memorization of noisy labels and the inferiority of sample-selection bias, which is very essential for extracting hard confident examples. (Hard confident examples are usually entangled with incorrectly labeled examples and could be extracted only when the memorization of noisy labels and the inferiority of sample-selection bias have been addressed well.)
>
> To further show the difference, we empirically compare Me-Momentum with SELF under the same setting. More details can be found in Appendix C.
>
> Dataset       SELF     Me-Momentum
> CIFAR10 Sym-40   87.35%    92.31%
> CIFAR10 Sym-60   75.47%    87.88%
> CIFAR100 Sym-40    61.40%    68.25%
> CIFAR100 Sym-60    50.60%    59.51%
>
> The results show that Me-Momentum largely outperforms SELF. Note that the backbone and loss function (cross-entropy loss function) are the same. The outperformance clearly shows that Me-Momentum can better extract high-quality confident examples significantly.

---

> ### Author Response · Authors · 2020-11-17
> **To AnonReviewer3 (1)**
>
> Thanks for the careful reading and nice comments!
>
> Q1: Running time comparison.
> A1: The number of the inner loop varies because the inner loop stops by exploiting a noisy validation set.
> Note that we discard non-confident examples and only use confident examples to train the model. The training time in each round would be much less than the normal training. We compare the training time with representative baselines on CIFAR10 with ResNet18 as follows:
>
> Methods        Training time
> CE           68 mins
> JointOptim        88 mins
> Co-teaching      91 mins
> T-revision       232 mins
> Me-Momentum     169.40±18.80 mins (symmetric 50%)
>               160.0±21.01 mins (instance 40%)
>
> Note that CE stands for the normal neural network training by employing the cross-entropy loss function. The total number of epochs for CE is set to 200. Since the number of the inner loop varies in the proposed method, we have reported the average training time of five runs and the standard deviation. Note that Me-momentum has a smaller training time on the instance-40% noise than that on the symmetric-50% noise. This may be caused by that the former setting is more difficult than the latter one and less confident examples are extracted.
>
> The conclusion is that the training time of the proposed method is shorter than T-revision but longer than Co-teaching, JointOptim, and CE.
>
>
> Q2: Do Tables 1-3 show a similar improvement to Table 4 when switching to clean validation set?
> Q2: Yes, they are. A clean validation set will help to select a better¬ model compared with the noisy validation set. The final performance will become better. The details are as follows (Each trial is repeated five times):
>
> Dataset       with noisy validation set   with clean validation set
> CIFAR10 Sym-20	  91.44±0.33          91.60±0.31
> CIFAR10 Sym-40	  88.39±0.34          89.14±0.51
> CIFAR10 Sym-50	  86.40±0.34          86.88±0.70
> CIFAR10 Inst-20 	   90.86±0.21          91.34±0.24
> CIFAR10 Inst-40    86.66±0.91          87.80±0.84
>
> Q3: How sensitive is Me-Momentum to hyperparameters N_outer and N_inner?
> A3: Thanks for the nice concern. We may not need to set specific values for N_outer and N_inner as we can stop the algorithm when there no improvement in the next inner or outer loop by looking at the validation set. However, this may be time-consuming. We set N_outer and N_inner to be specific values to save time.
>
> From Figure 3, we can see how the number of confident examples, confident label precision, and the test accuracy varies by increasing the number of the round of inner and outer loops in a limited range. We can see that the curves are quite smooth, which shows that the proposed method is not very sensitive to the hyperparameters.
>
> Q4: Is the recall of confident examples wrt accurately labeled data as high as the precision? Will identifying additional confident samples lead to even higher accuracy?
> A4: Thanks for the nice comment. We have listed the label precision and recall for the accurately labeled as follows:
>
> Dataset        Label precision  Recall
> CIFAR10 Sym-40    96.43±0.48     95.76±0.24
> CIFAR10 Inst-40     94.48±1.84     94.22±0.46
> CIFAR100 Sym-40  98.09±0.10     82.15±0.68
> CIFAR100 Inst-40   91.01±4.06     78.24±3.56
>
> We can see that for the dataset CIFAR10, the recall is as high as the precision, which justifies that we have extracted high-quality confident examples and why we have a good classification performance. However, the recall is relatively low compared with the precision on CIFAR100 where each class has a relatively small number, making it difficult to learn.
>
> Intuitively, with additional high-quality confident examples, the classification accuracy could be further improved. The proposed method aims to extract high-quality confident examples by justifying the proposed idea, i.e., momentum of memorization, without making use of the non-confident examples or a clean validation set. The accuracy could be improved by making use of the non-confident examples or a clean validation set, which will be studied in the future. Note that we will explain why we use the term “Momentum” later, which will provide the intuition why hard confident examples will be extracted and justify the novelty of the proposed method.

---

### Official Review · AnonReviewer4 · 2020-10-29
**Interesting Work**

**Rating:** 8
**Confidence:** 4

**Review:**

The authors introduce an interesting approach to handling hard "confident" samples in learning with label noises. At the heart of the proposed approach is an interactive method that jointly refines the classifier and the samples. The confident samples are initialized by utilizing the memorization effect of deep networks. Then, a classifier is learned from such samples.

During the learning process, hard confident data are selected progressively by looking at the classification results, which further better select confident samples.

Experimental results show that the proposed method achieves state of the art.

Pros:
1. The task studied here could be of interest to a large number of readers in the community.
2. The overall idea is quite interesting and intuitively makes sense. I like the fact that the core idea of the proposed approach finds its root in physics.
3. The results are very promising, in spite of the simple nature.
4. The manuscript is overall well-written and easy to follow.

Cons:
1. The iterative nature is good and bad. It seems to me that sometimes it requires many rounds in the inner loop, as shown in Fig. 3, for example, the number is up to 20 to 30. Please show some numbers in terms of running time and compare them with the baselines.
2. I might be missing something here but, what about the not-so-confident samples? Any scheme to take care of them?

Minor issues
1. The fonts in Fig. 3 should be enlarged.
2. It would be better if the author could elaborate more on its connection with the momentum in physics.

---

> ### Author Response · Authors · 2020-11-17
> **To AnonReviewer4**
>
> Thanks very much for the positive support!
>
> We address the concerns in cons one by one in the following.
>
> Q1: Running time comparison.
> A1: The number of the inner loop varies because the inner loop stops by exploiting a noisy validation set.
> Note that we discard non-confident examples and only use confident examples to train the model. The training time in each round would be much less than the normal training. We compare the training time with representative baselines on CIFAR10 with ResNet18 as follows:
>
> Methods     Training time
> CE        68 mins
> JointOptim     88 mins
> Co-teaching   91 mins
> T-revision    232 mins
> Me-Momentum  169.40 $\pm 18.80$ mins (symmetric 50%)
>            160.0 $\pm 21.01$ mins (instance 40%)
>
>
>
> Note that CE stands for the normal neural network training by employing the cross-entropy loss function. The total number of epochs for CE is set to 200. Since the number of the inner loop varies in the proposed method, we have reported the average training time of five runs and the standard deviation. Note that Me-momentum has a smaller training time on the instance-40% noise than that on the symmetric-50% noise. This may be caused by that the former setting is more difficult than the latter one and less confident examples are extracted.
>
> The conclusion is that the training time of the proposed method is shorter than T-revision but longer than Co-teaching, JointOptim, and CE.
>
> Q2: Regarding the not-so-confident examples.
> A2: Note that we have discarded the not-so-confident examples in each round. Our aim is to verify the effectiveness of the method to extract high-quality confident examples, not to boost the classification performance. The not-so-confident examples could be exploited, e.g., by the semi-supervised learning methods, to further improve the performance as did in DivideMix [r1].
>
> [r1] Li, Junnan, Richard Socher, and Steven CH Hoi. "DivideMix: Learning with Noisy Labels as Semi-supervised Learning." In ICLR. 2019.

---

### Decision · Program_Chairs · 2021-01-07
**Final Decision**

**Decision:**

Reject

**Comment:**

The authors propose a process to leverage the memorization effect of deep learning models to filter out examples at the boundary (hard) that the models are confident on, and argue that identifying those hard confident examples help improve the accuracy when learning under noisy data. The process essentially alternates between confident example selection and classifier updating, where the two parts are expected to help each other to form a positive cycle. Experiments demonstrate superior results over other self-purifying approaches.

The reviewers have a very diverse opinion about the paper. On the positive side, everyone agrees that the superior experimental results to be very impressive. The authors have addressed some concerns well, such as the running time. One reviewer pointed out that the current study has not been combined with semi-supervised learning yet, but during the discussion, most agreed that it is not a crucial negative point of the current paper. On the actual negative points, there are issues that were not cleared even after the rebuttal, such as whether re-initialization in the process helps escape local optimal, and the key difference between the "small-loss trick" and "memory-momentum trick." While the authors argued the novelty with respect to SELF, more illustrations and experiments are needed to highlight the novelty aspect.

Given the diverse opinions, the AC read the paper in detail, and assessed the reviewer's opinions and the authors rebuttal. Overall a serious concern is the leap of faith that the proposed process is indeed (a) "leveraging the memorization effect" to (b) "extract hard confident examples" to (c) "improve accuracy in noisy learning". For (a), it is mentioned that deep learning models "learns simple patterns on majority of data" first, where the majority in this work is argued is the clean ones. But there is no validation of this claim in the experiments. For instance, there is no figure/discussion that shows how much "clean data" has been correctly captured/memorized by the earlier deep learning models. For (b), the terminology of "hard but confident" is ill-defined. If examples being hard means them to be around the boundary, one can argue that they could never be "confident" as one measure of the confidence is the margin. The authors may want to mean "hard but clean", but then more illustrations are needed to analyze whether the extracted examples are really the clean ones, or if there are noisy ones being "confident" from the proposed process as well. For (c), it is then unclear whether the improved performance is caused by noise removal (as the authors hope to argue), or by just zooming in to the boundary (regardless of whether the extracted examples are clean or noisy). The authors are encouraged to not just look at the superior performance on accuracy, but analyze more on what actually happened behind the scenes to understand the proposed process better.

Some more comments from the AC that could help the authors (1) Are the examples kept by the proposed approach similar to the ones kept by SELF? Why or why not? Are there empirical studies on this? (2) For the competitors' approaches, what would their Figure 4 look like? In particular, for approaches that use "small-loss trick", what would Figure 4 look like? How would Figure 4 be different for the proposed process (i.e extracting hard "confident" examples) and others (like extracting just hard examples without considering confidence, or just confident examples without considering hardness)? (3) Are there studies that deliberately initialize the process with noisy data, to see how sensitive the process is before the "positive cycle" begins?